# Musculoskeletal Pain in Undergraduate Students Is Significantly Associated with Psychological Distress and Poor Sleep Quality

**DOI:** 10.3390/ijerph192113929

**Published:** 2022-10-26

**Authors:** Saad M. Alsaadi

**Affiliations:** College of Applied Medical Sciences, Imam Abdulrahman Bin Faisal University, Dammam 34212, Saudi Arabia; ssaadi@iau.edu.sa

**Keywords:** low back pain, musculoskeletal pain, neck pain, psychological distress, quality of life, sleep, undergraduates

## Abstract

Musculoskeletal pain (MSKP), psychological distress, and poor sleep quality are common among undergraduate university students. Yet, few studies have assessed the association between MSKP and psychological distress and poor sleep quality. This cross-sectional study was conducted to determine this association among undergraduate students at a major public university in Saudi Arabia. MSKP was assessed using the Nordic Musculoskeletal Questionnaire, psychological distress using the Depression, Anxiety and Stress (DASS-21) questionnaire, and sleep quality using the Pittsburgh Sleep Quality Index. A total of 339 undergraduate students from various specialties provided complete responses and were included. The most common site of MSKP in the past 12 months and the past 7 days were neck pain (54.6% and 41.9%, respectively) and low back pain (49.4% and 48.2%, respectively). There was no difference in the prevalence of MSKP across colleges. The reported MSKP in the past 12 months and 7 days were significantly associated with the students’ level of anxiety and stress as well as sleep quality (*p* < 0.05 for all), while depression was only significantly associated with MSKP in the past 7 days.

## 1. Introduction

Musculoskeletal pain (MSKP), defined as any pain or discomfort that affects bones, joints, ligaments, tendons, or muscles, can be caused by poor body ergonomics or certain repetitive movements [1,2]. MSKP affects all age groups, gender, and low- and high-income communities. According to the Global Burden of Diseases (1990–2019), of the top 10 drivers of the absolute disability-adjusted life years (DALYs), four conditions affect from teenage to older years, and these include low back pain and “other musculoskeletal disorders”. In fact, other musculoskeletal disorders had the second-highest increase (30.7%) of the top 10 largest contributors of additional DALYs over the 30-year period [3]. According to the World Health Organization (WHO), MSKP is also the leading cause of years lived with disability [4]. Indeed, MSKP can lead to a significant socioeconomic burden, including direct healthcare costs, work absenteeism, and psychological distress (i.e., depression, anxiety, and stress) [5]. 

University students worldwide have a sedentary lifestyle, spending an average of 10 h/day being sedentary [6]. A sedentary lifestyle predisposes to MSKP, and unsurprisingly, MSKP is common among university students [2]. Young adult undergraduate students also commonly suffer from sleep deprivation [7], and sleep deprivation has extensively been reported to aggravate MSKP as well as increase mental health-related risks [8]. Among university students, MSKP has been reported to result in poor academic performance and decreased leisure time [9,10]. 

Although the causal relationship between MSKP, sleep deprivation, and psychological distress is not entirely understood, the associative relationship is bidirectional. For example, while some studies have shown that a significant proportion of those with MSKP suffer from psychological distress, others have found psychological distress to be resulting in MSKP [11,12]. Similarly, low back pain, the most commonly reported MSKP, has been found to be significantly associated with the quality of sleep quality [13], and on the other hand, a night of poor sleep quality has been reported to be followed by worsened MSKP intensity [14]. Therefore, targeting the treatment of psychological and sleep elements to improve MSKP and its related disability has recently grown in importance [15,16]. 

Despite its importance, currently, there is limited available knowledge regarding the relationship between MSKP in university students and psychological factors and sleep quality. Therefore, the current study was conducted with the objective of determining the relationship between MSKP and psychological distress and sleep quality. The findings of this study would help formulate further interventions and recommendations toward alleviating MSKP in university students, and thus improve their quality of life and academic performance.

## 2. Materials and Methods

This study was prepared following the Strengthening the Reporting of Observational Studies in Epidemiology (STROBE) guidelines.

### 2.1. Study Design, Setting, and Participants

This cross-sectional self-administered questionnaire-based study included young adult undergraduate students who were enrolled at Imam Abdulrahman Bin Faisal University (IAU), Dammam, Saudi Arabia, during the academic year 2018–2019. The study was carried out between 1 March and 31 March 2019, after obtaining the ethical approval of the Institutional Review Board at IAU (Ref. no.: UGS-2019-03-077). IAU is one of the leading public universities of Saudi Arabia and the largest in the Eastern Province.

Undergraduate students at all 12 colleges of IAU were eligible for participation. The colleges were clustered into Health (comprising colleges of Medicine, Dentistry, Nursing, Applied Medical Sciences, and Public Health), Humanities (colleges of Arts and Education), Science (colleges of Basic Science and Business Administration), and Engineering (colleges of Engineering, Architecture, and Computer Sciences). The undergraduate degree duration of study ranged from 4 years (i.e., Basic Science, Business Administration, Arts, Education, Engineering, and Architecture) to 6 years (i.e., Medicine and Dentistry), with the study requirements and stress resulting in varying levels of strain on the musculoskeletal system.

Undergraduate students were invited to voluntarily participate in the study through posters that were placed at the entrances of all colleges of IAU. In addition, in-person approaches were made to students at random in the foyer of each college on a selected day. All prospective participants were initially screened for inclusion through a brief verbal interview. Students who reported trauma to any part of the body or surgical intervention in the past 12 months, physical disability, a clinical diagnosis of any mental disorder, or being on pain/sleep medications were excluded. 

A hard copy of a booklet comprising the questionnaire, a sheet explaining the purpose of the study, and a consent form was provided to the eligible prospective participants. Participants returned the completed questionnaire booklet through a designated box placed in each college. Participants were provided 1 week to respond to the questionnaire. 

All students were informed that participation was voluntary, and no incentives were offered. In addition, all participants signed a consent form prior to participation and were assured of anonymity and data protection. Further, the study adhered to the principles of the Declaration of Helsinki, 2013. 

### 2.2. Sample Size Calculation

The required sample size was calculated using Raosoft^®^ (http://www.raosoft.com/ accessed on 10 February 2019). IAU has about 23,000 undergraduate students across all colleges, and this was considered the study population. To maintain a confidence interval (CI) of 95% and a 5% margin of error, the sample size was calculated to be 378 participants. An additional 10% was added to the sample size to compensate for incomplete or no responses, and thus the final sample size required was 415 participants.

### 2.3. Outcomes

The primary outcomes were reporting the prevalence of each of the three parameters individually as well as assessing the association between MSKP, sleep deprivation, and psychological distress.

### 2.4. Study Tools

#### 2.4.1. Socio-Demographic Data

Data for socio-demographic variables such as age, gender, height, weight, marital status, and living with family/alone were collected using a questionnaire designed for this study. The other variables for which data were elicited using this questionnaire were smoking and caffeine consumption status and the self-reported GPA (range: 0–5).

GPA of 2.0–<2.75 (grading: 60–<70) was considered as fair, 2.75–<3.75 (grading: 70–<80) as good, 3.75–<4.50 (grading: 80–<90) as very good, and 4.50–5 (grading: 90–100) as excellent. Smoking was defined as those who smoked on a regular basis during the studied period.

#### 2.4.2. Musculoskeletal Pain Assessment

The validated Arabic version of the Nordic Musculoskeletal Questionnaire (NMQ) was used to determine the prevalence of MSKP among the participants [17]. The NMQ is a self-report questionnaire commonly used to detect musculoskeletal symptoms in an ergonomic or occupational health context. The NMQ questionnaire shows an image of the human body with questions divided into nine anatomical body regions (i.e., neck, shoulder, elbows, wrists and hands, upper back, lower back, hip, knee, and ankle and feet). The response to each question is binary, with “yes” and “no” indicating the presence and absence of MSKP, respectively [18]. Participants were asked to indicate whether they had an episode of pain or discomfort in the past 12 months and the past 7 days. Further, the questionnaire elicited data regarding the influence of pain/discomfort on daily life activities or visiting a physician in the past 12 months with a complaint of MSKP.

#### 2.4.3. Sleep Quality Assessment

The Arabic version of the Pittsburgh Sleep Quality Index (PSQI), which has been validated and found to be reliable, was used to evaluate students’ sleep quality [19]. PSQI is a self-report instrument designed to evaluate sleep quality in the past month. It consists of 19 items of the following seven components of sleep quality: sleep onset latency, sleep duration, efficiency, quality, disturbances, medication, and daytime dysfunction. Each component is scored between 0–3, and the sum of the seven components yields a global score of sleep quality (score range: 0–21); a high score is an indication of poor sleep quality. Scores >5 have been found to be an accurate cut-off score to distinguish between patients with primary insomnia and normal sleep [20].

#### 2.4.4. Psychological Distress Assessment

Psychological distress was assessed using the validated Arabic version of the Depression, Anxiety, and Stress (DASS-21) questionnaire [21]. The DASS-21 is a self-report scale developed to assess the negative emotional states of depression, anxiety, and stress. Each psychological state is assessed through seven items. The depression-related items assess dysphoria, hopelessness, devaluation of life, self-deprecation, lack of interest/involvement, anhedonia, and inertia. The anxiety-related items assess autonomic arousal, skeletal muscle effects, situational anxiety, and subjective experience of anxious affect. The stress-related items assess difficulty relaxing, nervous arousal, and being easily upset/agitated, irritable/over-reactive, and impatient. Scores for depression, anxiety, and stress are calculated by summing the scores for the relevant items, and scores of more than 9, 7, and 14 are considered abnormal, respectively [22].

### 2.5. Statistical Analysis

Data were analyzed using SPSS 25 for Windows (IBM Corp., Armonk, NY, USA). Frequency and percentage were calculated for categorical variables, while mean and standard deviation were expressed as continuous variables in descriptive statistics. A minimum of 80% of all questionnaires were required to be completed for inclusion in the final analyses. The item mean imputation method was used to account for missing values while calculating the mean. The frequency of MSKP in the past week and past 12 months for each of the body sites was obtained, and the most frequent sites of pain were used in regression analysis to explore associated factors. The Chi-square test was used to obtain the odds ratio (OR) and 95% CI for the relation between categorical dependent and independent variables. For categorical dependent and continuous independent variables, the univariate binary logistic regression analysis was conducted.

## 3. Results

### 3.1. Demographics

A total of 400 students agreed to participate, of which 357 (89.3%) provided a response. However, 18 responses were excluded because they did not meet the minimum questionnaire completion requirement of the study. Therefore, the responses of 339 participants were included in the final analysis. The mean age of the participants was 21 ± 1.7 years (range: 18–31 years), and the majority were female (279; 82%). In terms of college-wise distribution, the majority were from health colleges (181; 53.4%) followed by humanities colleges (123; 36.3%); the fewest were from engineering colleges (7; 2%) (Table 1).

### 3.2. Psychological Distress

The mean scores of the DASS-21 indicated mild depression and stress (13.4 ± 10.4 and 14.9 ± 10.4, respectively) and moderate anxiety (11.1 ± 8.8) among the participants (Table 2).

### 3.3. Sleep Quality Assessment

The mean PSQI score was 8.8 ± 3.1, with most (85%) students having poor sleep quality (i.e., an overall PSQI score > 5) (Table 2). The highest proportion of participants studying humanities had poor sleep quality (93%), while the lowest proportion was in health sciences (80%). Sleep quality was not associated with students’ academic performance, but the domain of specialization was significantly associated with poor sleep quality (*p* = 0.01). 

### 3.4. Musculoskeletal Pain Assessment

Neck pain and low back pain were the most common regions of MSKP. More than half of the participants (54.6%) reported having neck pain in the past 12 months, while 41.9% reported having it in the past 7 days. Similarly, 49.4% and 48.2% reported having low back pain in the past 12 months and 7 days, respectively (Table 3). The prevalence of MSKP did not differ based on the colleges (Figure 1 and Figure 2).

### 3.5. Association between Musculoskeletal Pain and Psychological Distress and Sleep Quality

MSKP was significantly associated with psychological distress and poor sleep quality among the respondents (*p* < 0.05). Specifically, anxiety and stress were significantly associated with the reported MSKP in both the past 12 months and 7 days, while depression symptoms were significantly associated only with reported MSKP in the past 7 days. Sleep quality was significantly associated with reported MSKP in the last 12 months (*p* = 0.04) and 7 days (*p* < 0.001). On the other hand, there was no significant association between the reported MSKP and the demographic data, GPA, caffeine intake, or smoking habit (Table 4 and Table 5).

## 4. Discussion

This study found that neck and low back pain affected about half of the undergraduate students at a major public university in Saudi Arabia both in the past 12 months and the past 7 days, and that this did not differ based on the study specialty. In addition, MSKP in the past 12 months and 7 days were significantly associated with the students’ level of anxiety and stress as well as sleep quality, while MSKP in the past 7 days was also significantly associated with depression. It should be noted that although females accounted for most respondents in our study, this was closely representative of the male: female population distribution at our university. 

In terms of the prevalence of MSKP, the findings of the current study are coherent with those of a recent systematic review that found similar MSKP prevalence among university students in the past 12 months and 7 days as well as found neck pain and low back pain to be the most common sites of pain [2]. Similarly, in a recent study from Saudi Arabia that included undergraduate dental students, neck pain and low back pain were found to be the most prevalent sites of MSKP. The 12-month neck and low back pain prevalence rates of that study (69% and 65%, respectively) were higher than those of the present study, but this was expected given that dental students have relatively higher MSKP levels owing to their posture during practice [23]. Previous studies have found that females were more likely to suffer from MSKP than males, which was suggested to be due to their smaller body dimensions and lower muscle endurance [23,24,25,26]; however, this was not the case in our study, as MSKP equally affected about half of the male and female respondents at both analyzed timeframes. 

Pain and psychological distress are among the leading causes of quality-adjusted life years in young adults [3]. The current study found a significant association between MSKP and stress and anxiety at all time-point assessments. In terms of stress, our findings contrast that of another study from Saudi Arabia that did not find a significant association between stress and MSKP among medical students [27]. In terms of anxiety, the finding of the current study is in line with that of a longitudinal study from Sweden, which included students and found that anxiety was significantly associated with MSKP and was a predictor of pain chronicity [28]. Compared with the general population, several factors predispose the student population to stress, such as the lack of leisure time and financial requirements, and anxiety, such as examinations and academic assignments [9,27,28]. Psychological distress, in turn, can lead to poor academic performance and muscular tension, consequently resulting in MSKP [29,30]. Nonetheless, the association does not infer causation, and thus it is yet unclear if the reported MSKP among university students is caused by bad ergonomics and/or psychological factors such as stress/anxiety. 

A significant association was noted between depression and MSKP in the past 7 days, but not in the past 12 months. In general, the association between pain and depression is well recognized, with about half or more of those with pain having depression and vice versa [31,32,33]. A recent systematic review also highlighted that students with chronic pain (i.e., of a duration of >3 months) were more likely to report depression and poorer functioning than those without chronic pain [34]. However, to the best of our knowledge, there is a lack of literature wherein depression is significantly associated with MSKP pain in the past 7 days but not 12 months, and thus further studies are warranted to explore the reasons for this discrepancy. 

The present study found sleep quality was associated with MSKP both in the past 12 months and 7 days. This finding is in accord with those of our previous study, in which a strong relationship was noted between sleep quality and low back pain [13], and other studies that included undergraduate students and adolescents [35,36]. Moreover, the relationship between sleep quality and intensity of pain is likely bidirectional, i.e., a night of poor sleep quality is followed by a day with increased severe pain intensity and vice versa [14]. The current study did not find an association between sleep quality and academic performance. Similarly, Yuar et al., in their study that included 290 medical students and also used PSOI, found that 98% of the respondents had poor sleep quality but this did not affect their academic performance [37]. 

### 4.1. Study Implications 

The study found a high prevalence of MSKP, and thus there is a need for intervention from policymakers to undertake corrective measures to improve the quality of life of undergraduate students in Saudi Arabia. The findings of this study also highlight the need for providing guidance and support to undergraduate students suffering from MSKP and psychological distress; behavioral modification techniques have been found to be effective in the management of depression and anxiety in patients with MSKP [11]. The university information technology departments could collaborate with different departments to create a telemedicine center to provide care for students suffering from MSKP and/or psychological distress. It has several advantages, including increased treatment adherence, low cost, and reduced load on university clinics [11].

### 4.2. Limitations

The current study has several limitations that should be considered while interpreting its results. Randomly approaching students to participate may have potentially introduced selection bias. In addition, the study did not record several factors that can have a confounding effect on the studied parameters, such as the usage of electronic devices such as mobile phones, the amount of leisure time, level of athletic or physical activity [38,39], and the students’ current level of study (i.e., years in college) [40]. The study reports the collective data from various colleges, and thus assumes parity in the stress levels (physical and mental) between all colleges; however, this is unlikely to be true, and future studies could be conducted only in a specific college or similar course colleges for comparison. The study also did not determine the timepoint in the past 12 months when MSKP occurred, which may have correlated with stressful academic events such as examinations, and this should be measured in future studies to obtain a better understanding of factors leading to the studied parameters. The study design does not allow inference of whether the reported MSKP is a cause for psychological distress or vice versa. Therefore, additional studies are required to overcome these limitations and better identify the direction of the association. Finally, the proportion of females in the current study was extremely high and this may have affected certain analyses; for example, smokers accounted for about 10% of the respondents, which is lower than the general trends in Saudi Arabia, and this may be due reporting bias by the female participants owing to social norms, which in turn is likely to have affected the association analysis. 

## 5. Conclusions

This study found that, in undergraduate students at a major public university from Saudi Arabia, neck and low back pain are the most common MSKP in the past 12 months and 7 days, and most students have poor sleep quality. Anxiety and stress as well as poor sleep quality were significantly associated with MSKP in both the past 12 months and 7 days. The findings of this study serve as baseline data that indicate an absolute need for similar prospective, case-control/randomized studies. The findings from such a study would help formulate interventions and recommendations for alleviating MSKP and improving the psychological condition and sleep quality in university students, which in turn would likely improve their quality of life.

## Figures and Tables

**Figure 1 ijerph-19-13929-f001:**
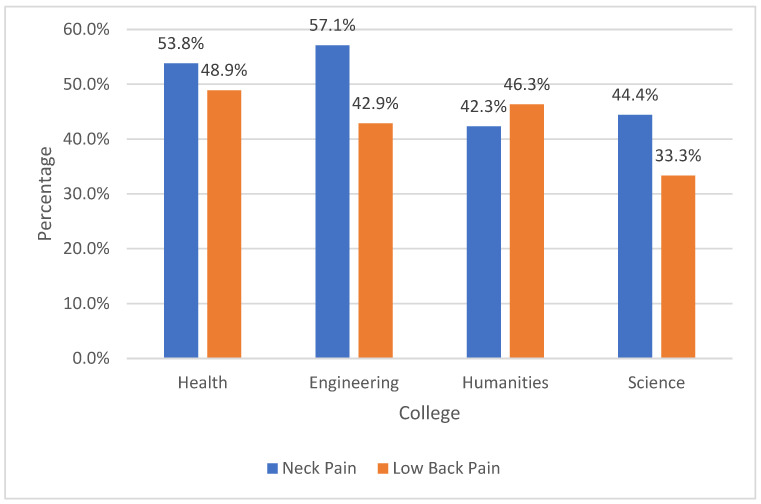
Prevalence of MSKP among university colleges in the past 12 months.

**Figure 2 ijerph-19-13929-f002:**
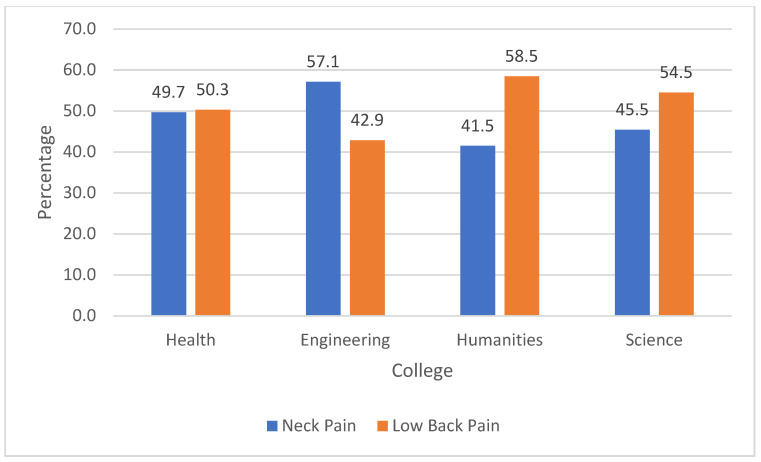
Prevalence of MSKP among university colleges last 7 days.

**Table 1 ijerph-19-13929-t001:** Demographic information of the participants.

Demographic	*n*
Age (years) (mean ± SD)	21 (±1.7)
Gender (Female)	279 (82%)
BMI (kg/m^2^) (mean ± SD)	22.3 (±4.6)
Caffeine intake (cup/day) (mean ± SD)	1.0 (±0.89)
Smoker	
Yes	37 (11%)
No	302 (89%)
Marital status	
Single	300 (88.5%)
Married	39 (11.5%)
Divorced/Widowed	0
Live alone (No)	311 (91.7%)
Specialty	
Health	182 (54%)
Humanitarian	123 (36%)
Science	27 (8%)
Engineering	7 (2%)
GPA (0–5)	
Fair	5 (1.5%)
Good	149 (44.2%)
Very good	115 (34.1%)
Excellent	68 (20.2%)

**Table 2 ijerph-19-13929-t002:** Psychological distress and sleep quality scores.

Parameter	Mean ± SD
Psychological distress	
Depression	13.4 ± 10.4
Anxiety	11.1 ± 8.8
Stress	14.9 ± 10.4
Sleep quality (PSQI score)	8.8 ± 3.1

PSQI—Pittsburgh Sleep Quality Index.

**Table 3 ijerph-19-13929-t003:** Prevalence of MSKP in the past 12 months and 7 days.

Pain Area	During the Past 12 Months *n* (%)	During the Past 7 Days *n* (%)	Preventing Function *n* (%)	Seeing Physician for Pain *n* (%)
Neck	185 (54.6)	142 (41.9)	51 (16.1)	18 (5.6)
shoulders	143 (44.7)	120 (368)	54 (17.1)	23 (7.3)
Upper back	113 (35.5)	98 (30.2)	52 (16.6)	24 (7.5)
Elbow	24 (7.6)	32 (9.9)	24 (7.6)	19 (6.0)
Wrist/hands	67 (21.2)	61 (18.8)	41 (12.9)	18 (5.7)
Lower back	158 (49.4)	163 (48.2)	73 (22.7)	32 (10.1)
Hip/thigh	65 (20.4)	59 (18.2)	36 (11.4)	23 (7.3)
knees	68 (21.2)	71 (21.8)	35 (11.0)	24 (7.5)
Ankle/feet	77 (24.1)	69 (21.2)	43 (13.5)	32 (10.1)

**Table 4 ijerph-19-13929-t004:** Factors associated with reported MSKP pain in the past 12 months.

Variable	MSK Pain		Statistic Test	*p*-Value
	Yes	No		
Gender				
Male	28	32	0.04	0.83
Female	126	153		
Smoke				
Yes	22	14	2.23	0.13
No	136	148		
College				
Health	89	89		
Humanities	57	52		
Science	18	9	3.1	0.36
Engineering	3	3		
GPA				
Fair	1	4		
Good	72	68		
Very good	54	58	2	0.56
Excellent	31	32		
Caffeine intake (cup/day)				
0	44	57		
1–2	79	67		
3–4	24	21	6.96	0.13
>4	8	15		
	Statistic test	*p*-value		
Age	0.06	0.42		
Depression	0.01	0.49		
Anxiety	0.05	0.002		
Stress	0.03	0.03		
Sleep Quality	0.1	0.04		

**Table 5 ijerph-19-13929-t005:** Factors associated with reported MSKP in the past 7 days.

Variable	MSK Pain		Statistic Test	*p*-Value
	Yes	No		
Gender			0.29	0.58
Male	36	23		
Female	159	119		
Smoke			2.1	0.14
Yes	22	15		
No	141	160		
College			0.72	0.86
Health	85	96		
Humanities	62	61		
Science	12	15		
Engineering	4	3		
GPA			5.12	0.16
Fair	1	4		
Good	75	74		
Very good	49	67		
Excellent	38	30		
Caffeine intake (cup/day)			5.06	0.28
0	48	54		
1–2	78	80		
3–4	24	23		
>4	2	17		
	Statistic test	*p*-value		
Age	0.09	0.15		
Depression	0.03	0.004		
Anxiety	0.05	<0.001		
Stress	0.05	<0.001		
Sleep Quality	0.03	<0.001		

## Data Availability

The data presented in this study are available on request from the corresponding author. The data are not publicly available due to a requirement of the IRB.

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
