# Peer review of "Musculoskeletal Pain in Undergraduate Students Is Significantly Associated with Psychological Distress and Poor Sleep Quality"

_ijerph, 2022, doi:10.3390/ijerph192113929_

Round 1

Reviewer 1 Report

  • Thank you for the opportunity to review this interesting paper. Below are some comments that should be addressed:
  • Line 29: What does it mean two of the four top 10?
  • Lines 63-68: First, study design must be explained. Second, setting and study population must be explained in detail. What colleges are there and how many? Describe the studies - how many years are there (how long the studies are etc.). Are the studies different in any way? Some studies might put less strain on the students in terms of stress, burnout, even MSKP. All of this must be elaborated.
  • Lines 70-71: Randomly approaching students to participate introduce selection bias. Why was this approach chosen? This must be acknowledged in the limitations.
  • Lines 71-75: How did you check for inclusion and exclusion criteria? For example, did you verify the absence of a diagnosis of any mental disorder or was this self-reported? Add explanations.
  • Lines 85-86: Sample size should be explained more specifically. Based on which previous study was the sample size calculated (data on prevalence)? What was the size of population from which the sample was derived? Why was a CI of 90% chosen when 95 or even 99% are common choices? Was a software, and which if yes, used to do calculations? All of this information is important and must be clarified and explained.
  • Lines 90-92: Shouldn't the first outcome of interest be prevalence of each, and then the association between these outcomes?
  • Line 93: Apart from the three specific questionnaires, describe the one used to obtain socio-demographic characteristics of participants - was it constructed specifically for this study, what questions did it contain etc.?
  • Line 93: For all three used questionnaires, in which language were they administered? Provide details and citations of validation studies in the administered language for each of the three questionnaires.
  • Lines 131-132: How can 80% of completed questions be considered as a complete questionnaire? How were outcomes calculated for these students if answers were missing? How was missing data handled in the analysis?
  • Methods: Report participation rate and response rate.
  • Table 1 needs to be revised. What does it mean that caffeine intake was 1.0 (0.89%) - only one students consumed coffee?
  • Table 1: What were possible answers for some of the questions - e.g. marital status (which is repetitive - were there only two answers offered, were students asked about being in a relationship, divorced, widows), caffeine intake - list all answers and frequencies per each.
  • Table 1: In methods, define GPA classification. Also, describe the grading system at the University - these differ across the countries worldwide, and reader needs to be informed.
  • Table 1: DASS and PSQI score results are not demographic characteristics - revise and consider reconstructing this table.
  • Lines 155-157: How come reported MSKP did not lead students to seek medical care when Table 2 has an entire column describing how many students had to see a physician for pain?
  • Table 2: Word pain is missing in the title.
  • Table 2: Check the percentage for hip/thigh - 22.6.2.
  • Table 2: If 339 participants were included in the analysis, how come 168 is 50%? Check the percentages throughout the Tables.
  • Figure 1: Title should not be within the figure, since there is a caption below. And explanation for orange colour is missing. Also, abbreviations need to be explained.
  • Results: What did you adjust for in your analyses? Did you check BMI and included it in your models?
  • Line 215: Should you compare results of a sample of students aged 18-31 with 16-year old secondary school students? Compare results with similar studies conducted in similar populations.
  • Lines 224-232: This paragraph is inconsistent. You found a significant association between MSKP in the past 7 days and depression, but not for MSKP in the past 12 months. Then you state that your findings are consistent with a systematic review that explains how chronic pain (is this 7 days?) is associated with depression. Then again you state how there are no studies for your results. Revise.
  • Discussion: This section needs to be revised. More comparisons with studies done in similar populations need to be included - and in particular in terms of different colleges that were included - you stated that no difference was found between colleges, yet mostly the section discussion includes comparison with studies done in medical students. Revise, expand the comparisons. Explore potential reasons for observed differences.
  • Conclusions: This should not be a simple summary of your findings but emphasis on the most important findings and in particular their implications - for students, for authorities, for research - this must be elaborated briefly here.
  • There are several English grammar mistakes that should be corrected throughout the manuscript.

Author Response

Dear Reviewer 1, 

I would like to sincerely thank you for having dedicated precious time in reviewing this manuscript and providing in-depth comments on the manuscript. I firmly believe that these comments have tremendously helped in raising the overall quality of the manuscript. 

Please find my point-by-point responses at the end of the attached file (along with the yellow highlights indicating the changes in the main body of manuscript).

Best regards, 

Reviewer 2 Report

First of all, thank you for being able to review this work.

Congratulations to the author for his efforts.

I will now comment:

I am of the opinion that it is biased not to comment: The type of study carried out, at what time the pain appears, i.e. exam time; the university study carried out.

In relation to the sample, the CONSORT methodology must be followed.

The tables presented are not necessary.

The mean age of the participants 144 was 21 (±1.7) years (range: 18–31 years). 

  • Submit it together
  • Appears on line 170,181 (P = ..)??
  •  

Author Response

Dear Reviewer 2, 

I would like to sincerely thank you for having dedicated precious time in reviewing this manuscript and providing valued comments on the manuscript. I firmly believe that these comments have helped improve overall quality of the manuscript. 

Please find my point-by-point responses at the end of the attached file (along with the yellow highlights indicating the changes in the main body of manuscript).

Best regards, 

Round 2

Reviewer 1 Report

I would like to thank the Author for carefully revising their manuscript and incorporating relevant changes. The manuscript looks much better now. Please see minor comments below:  
  • Methods: This study instead of this manuscript was done according to STROBE.
  • Methods: Was the screening interview for inclusion verbal or written? Specify.
  • Table 1: Check the column for "n" - should the (%) be omitted?

Author Response

Dear Reviewer 1, 

We are thankful to your valued comments and kind words on our revised manuscript. 

We have further revised the manuscript based on your comment. 

Please find our responses as follows:

Reviewer's comment: "Methods: This study instead of this manuscript was done according to STROBE."

Author's Response: We have replaced "manuscript" with "study" 

Reviewer's comment: "Methods: Was the screening interview for inclusion verbal or written? Specify."

Author's Response: The interview was verbal; we have now added the same. 

Reviewer's comment: Table 1: Check the column for "n" - should the (%) be omitted?

Author's Response: We have deleted "(%)" from the column head and added "%" throughout the table at the applicable places. 

We hope you find the above responses to be satisfactory. 

Kind regards, 

Reviewer 2 Report

Congratulations on your efforts;

Just correct reference 42 in the bibliography.

Author Response

Dear Reviewer 2, 

Thank you for your valued comment throughout the peer review process of this manuscript and for your kind words on the revised manuscript. 

Per your comment, we have now deleted the extra "42" in Reference 42. 

We hope you find all corrections in the manuscript to be satisfactory. 

Kind regards,